# Relative Representations:
# Topological and Geometric Perspectives

**Alejandro García-Castellanos** [*]
Amsterdam Machine Learning Lab
University of Amsterdam, Netherlands
`a.garciacastellanos@uva.nl`

**Giovanni Luca Marchetti**
Department of Mathematics
KTH Royal Institute of Technology, Sweden

**Danica Kragic**
Division of Robotics, Perception and Learning
KTH Royal Institute of Technology, Sweden

**Martina Scolamiero**
Department of Mathematics
KTH Royal Institute of Technology, Sweden

## Abstract

Relative representations are an established approach to zero-shot model stitching, consisting of a non-trainable transformation of the latent space of a deep neural network. Based on insights of topological and geometric nature, we propose two improvements to relative representations. First, we introduce a normalization procedure in the relative transformation, resulting in invariance to non-isotropic rescalings and permutations. The latter coincides with the symmetries in parameter space induced by common activation functions. Second, we propose to deploy topological densification when fine-tuning relative representations, a topological regularization loss encouraging clustering within classes. We provide an empirical investigation on a natural language task, where both the proposed variations yield improved performance on zero-shot model stitching.

## 1 Introduction and Related Work

The ability to infer semantically-rich representations is, perhaps, the cornerstone of the success of contemporary deep learning models [4]. Latent spaces of deep neural networks extract features from data that are general and transferable, meaning that, after training a model on a given task, its representations can be leveraged upon for addressing a variety of related tasks. This principle is nowadays exploited in the form of *foundation models* [5] – neural networks trained via self-supervision on large-scale multi-task datasets to extract ready-to-use representations.

Surprisingly, it has been argued that neural networks trained on diverse tasks, architectures, and domains, infer structurally similar representations [34, 25, 38] – a hypothesis sometimes referred to as 'representational universality' or 'convergent learning' [29]. For example, there is extensive empirical evidence that the representations extracted by neural networks are isometric (up to scale) – i.e., are related by an affine distance-preserving transformation – as the hyperparameters and initialization vary [35]. Even though partial theoretical explanations have been proposed – e.g., based on harmonic analysis [32, 11] or on philosophical principles [22] – these phenomena remain mostly mysterious. Nonetheless, they have motivated the introduction of techniques for zero-shot transfer and latent space communication, such as *model stitching* [28, 2, 12] – a method consisting in a trainable layer that connects latent representations of two different networks. On a similar note, *relative representations* [35, 7] involve a non-trainable isometry-invariant layer on top of the representation. Assuming the

---

[*]Work done at the Division of Robotics, Perception, and Learning at KTH Royal Institute of Technology, Sweden.

representational universality hypothesis, factoring out rigid transformations results in features that are independent of nuances in initialization, hyperparameters, and, to some extent, task and architecture.

In this work, we propose two improvements of relative representations, enhancing the resulting zero-shot model stitching. These improvements are inspired by insights from geometry and topology, respectively.

**Geometric Perspective.** From the geometric side, we consider symmetries of neural networks [18, 15], i.e., transformations of the weights that do not alter the function defined by the network. These transformations are sometimes referred to as *intertwiners* [17], and form a group that depends on the activation function. We argue that these symmetries are partially responsible for the universality of representations with respect to initialization and training noise, and consequently design a relative transformation that is invariant to the intertwiner group of common activations. Our idea is simple: we propose to deploy batch normalization before the relative representation layer. This factors out non-isotropic rescalings, which are the non-isometric transformations induced by intertwiner groups for a wide class of activation functions.

**Topological Perspective.** From the topological side, we draw inspiration from topological data analysis and, in particular, from methods for *topological regularization* [33, 21, 20, 10, 37, 9]. The latter aims at enforcing specific topological features in the latent representation of a neural network. To this end, a recently-introduced method deemed topological *densification* [21] forces data classes to be represented in compressed clusters, resulting in a representation that is coherent with the decision boundaries of the neural network. We propose to deploy topological densification in conjunction with relative representations, and explore various alternatives for combining the two. Our intuition is that consistent densified representations share a similar topology, and are therefore more universal, while still preserving generality and transferability. Moreover, topological regularization prevents potential overfitting of large pre-trained models – such as foundation models – when fine-tuning them via relative transformations in low data regimes.

We implement and validate empirically both of the above proposals in an experimental scenario similar to the original work [35]. The scenario consists of a natural language task, where domains correspond to different languages, and model stitching enables zero-shot translation. Results show that both our variant of the relative transformation and the additional topological densification result in significantly improved performance with respect to the original version. In summary, our contributions are:

- A novel normalized variant of the relative representation that is invariant to the intertwiner group of common activation functions.

- The introduction of topological densification for fine-tuning relative representations.

- An empirical investigation on a natural language task, showcasing improved performance.

## 2 Background

### 2.1 Relative Representations

In this section, we overview *relative representations* [35] – a technique enabling zero-shot model stitching. The core idea is introducing a transformation in latent spaces that increases compatibility, without the need for training additional components. As explained in Section 1, the approach is based on empirical evidence that latent representations inferred by neural networks are, usually, close to being isometric (up to scale).

Let $\varphi\colon \mathcal{X} \to \mathcal{Z}$ be a representation, i.e., a map encoding data in $\mathcal{X}$ to a latent space $\mathcal{Z}$. Moreover, let $\mathcal{A} = \{a_1, \ldots, a_k\} \subset \mathcal{Z}$ be a set whose elements are referred to as *anchors*, and let $\mathrm{sim}\colon \mathcal{Z} \times \mathcal{Z} \to \mathbb{R}$ be a function representing a measure of similarity on $\mathcal{Z}$.

**Definition 2.1** ([35]). The *relative representation* of $z \in \mathcal{Z}$ w.r.t. $\mathcal{A}$ is

$$T_{\mathrm{rel}}^{\mathcal{A}}(z) = (\mathrm{sim}(z, a_1), \ldots, \mathrm{sim}(z, a_k)) \in \mathbb{R}^k .$$

If $\mathcal{Z} = \mathbb{R}^m$ and $\text{sim} = z \cdot z'/\|z\|\|z'\|$ is cosine similarity, then $T_{\text{rel}}$ is invariant to linear isometries and rescalings. More precisely, if $z$ and all the anchors in $\mathcal{A}$ are transformed via $z \mapsto \alpha U z$ – where $\alpha \in \mathbb{R}$, and $U$ is an orthogonal $m \times m$ matrix – obtaining $z'$ and $\mathcal{A}'$, then $T_{\text{rel}}^{\mathcal{A}}(z) = T_{\text{rel}}^{\mathcal{A}'}(z')$.

Relative representations enable zero-shot model stitching as follows. If $f_1 = \gamma_1 \circ \varphi_1$ and $f_2 = \gamma_2 \circ \varphi_2$ are two neural networks with the same architecture decomposed in a representation map $\varphi_\bullet$ and a head $\gamma_\bullet$, then one can replace $\gamma_1$ with $\gamma_2 \circ T_{\text{rel}}^{\mathcal{A}}$ to exploit the representation inferred by $f_1$ in conjunction with the head of $f_2$, and vice versa.

Assuming the representational universality hypothesis, this model stitching procedure extends to *cross-domain* setups, i.e., for two networks $f_1$ and $f_2$ that are trained over two distinct datasets from the same semantic domain (see Figure 1). In this case, to obtain coherent anchors, we fix $\mathcal{A}$ as an encoding via $\varphi_1$ of some data, and exploit a known correspondence between the datasets – based on domain knowledge – to obtain anchors $\mathcal{A}^*$ as encodings via $\varphi_2$ of the corresponding data. Lastly, for our purposes it will be convenient to resort to an end-to-end training setup, similarly to [27]. This consists in deploying pretrained representation maps $\varphi_1, \varphi_2$, and training

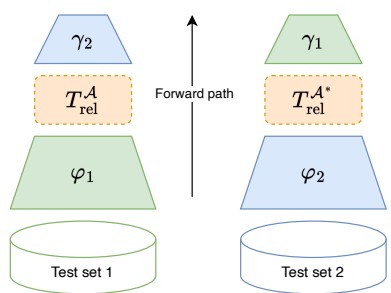

Figure 1: Cross-domain model stitching.

the networks $\gamma_1 \circ T_{\text{rel}}^{\mathcal{A}} \circ \varphi_1$ and $\gamma_2 \circ T_{\text{rel}}^{\mathcal{A}^*} \circ \varphi_2$ with the weights of both representation maps unfrozen, using their corresponding domain-specific dataset.

## 2.2 Symmetry Groups of Activation Functions

In this section, we overview a recent work that investigates the symmetries in neural networks [17]. The central idea is that certain activation functions induce similar symmetries in both weight space and latent representations. These symmetries generate the so-called *intertwiner group*.

Let $\text{GL}_n$ be the groups of $n \times n$ invertible matrices and $\sigma \colon \mathbb{R} \to \mathbb{R}$ a function. The latter represents the activation function of a neural network, and we will consequently extend it coordinate-wise as a map $\sigma \colon \mathbb{R}^n \to \mathbb{R}^n$.

**Definition 2.2.** The *intertwiner group* of $\sigma$ is:

$$G_\sigma^n = \{A \in \text{GL}_n \mid \exists B \in \text{GL}_n \colon \sigma \circ A = B \circ \sigma\}.$$

Suppose that $\sigma(I_n)$ is invertible, and for each $A \in \text{GL}_n$ define $\lambda_\sigma(A) = \sigma(A)\sigma(I_n)^{-1}$. It can be shown that, under the mild condition on $\sigma$, $G_\sigma^n$ is a subgroup of $\text{GL}_n$ and $\lambda_\sigma \colon G_\sigma^n \to \text{GL}_n$ is a homomorphism such that $\sigma \circ A = \lambda_\sigma(A) \circ \sigma$. Moreover, for common activation functions such as ReLU, GELU, and sigmoid, all the elements of $G_\sigma^n$ can be decomposed as a product of a permutation matrix and a diagonal one [17]. Since permutation matrices are isometries and isotropic diagonal matrices are (non-isotropic) rescalings, this draws a connection with the transformations factored out by the relative representations from Section 2.1.

The following elementary result shows that symmetries from the intertwiner group induce symmetries in the latent representations of a deep neural network. Let $f(x, W)$ be a multi-layer perceptron with input $x$, activation function $\sigma$, and layer-wise weights $W = (W_1, b_1, \ldots, W_l, b_l)$ with $W_i \in \mathbb{R}^{n_i \times n_{i-1}}$ and $b_i \in \mathbb{R}^{n_i}$. For each $1 \leq m \leq l$, consider the decomposition $f = \gamma_m \circ \varphi_m$ into the latent representation at the $m$-th layer and the corresponding head.

**Proposition 2.1** ([17]). *For each $1 \leq i < l$ pick $A_i \in G_\sigma^{n_i}$, and consider*

$$\widetilde{W} = (A_1 W_1, A_1 b_1, A_2 W_2 \lambda_\sigma(A_1^{-1}), A_2 b_2, \ldots, W_l \lambda_\sigma(A_{l-1}^{-1}), b_l).$$

*Then for each $m$:*

$$\varphi_m(x, \widetilde{W}) = \lambda_\sigma(A_m) \circ \varphi_m(x, W),$$
$$\gamma_m(x, \widetilde{W}) = \gamma_m(x, W) \circ \lambda_\sigma(A_m)^{-1}.$$

*In particular, $f(x, \widetilde{W}) = f(x, W)$ for all $x \in \mathbb{R}^{n_0}$.*

The above symmetries provide a theoretical explanation for the emergence of structurally-similar representations in networks with different initializations.

## 2.3 Topological Densification

In this section, we recall the basics of hierarchical clustering, and review the topological densification method from [21]. To this end, let $\mathcal{X}$ be a metric space with distance function $d \colon \mathcal{X} \times \mathcal{X} \to \mathbb{R}_{\geq 0}$, and $\mathcal{D} \subset \mathcal{X}$ be a finite subset. In practice, $\mathcal{D}$ will represent a dataset in an ambient space $\mathcal{X}$, or in a representation.

**Definition 2.3.** Given $\varepsilon \in \mathbb{R}_{>0}$, the *truncation graph* $\Gamma_\varepsilon(\mathcal{D})$ is the finite undirected graph with elements of $\mathcal{D}$ as vertices and an edge between $x, y \in \mathcal{D}$ if, and only if, $d(x, y) < \varepsilon$.

Connected components of the truncation graph can be interpreted as clusters of data. This is the idea behind density-based spatial clustering methods [14]. Instead, *hierarchical* density-based clustering [6] – as well as the related notion of $0$-th persistent homology – considers the evolution of clusters as the parameter $\varepsilon$ varies. For $\varepsilon = 0$ all the points in $\mathcal{D}$ form a distinct connected component. As $\varepsilon$ grows, edges are added to the truncation graph. This implies that if $\varepsilon \leq \delta$, each component in $\Gamma_\varepsilon(\mathcal{D})$ is contained in a component of $\Gamma_\delta(\mathcal{D})$. Moreover, two distinct components in $\Gamma_\varepsilon(\mathcal{D})$ can merge together and become connected in $\Gamma_\delta(\mathcal{D})$. The value of $\varepsilon$ at which two connected components merge is called a *death time*. Since more than two connected components can merge at the same parameter value, death times form a multi-set in $\mathbb{R}_{\geq 0}$ with multiplicity given by the number of merged components minus one. This multi-set is denoted by $\dagger(\mathcal{D})$, and it is equivalent to the *persistence diagram* of $0$-dimensional persistent homology. It can be shown that death times coincide exactly with the lengths of the edges of the mini-

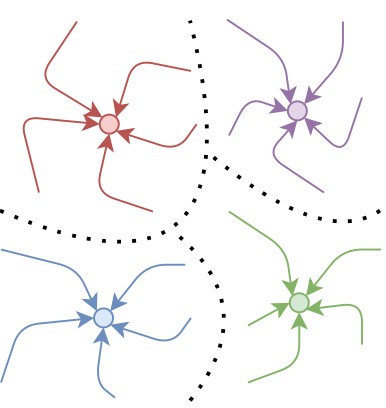

Figure 2: Effect of topological densification.

mum spanning tree of $\mathcal{D}$, i.e., the connected tree with vertices $\mathcal{D}$ and that is of minimal total edge length. This enables to compute the persistence diagram via, for example, Kruskal's algorithm [26], whose time complexity is $\mathcal{O}(|\mathcal{D}|^2 \log |\mathcal{D}|)$.

The concept of connectivity presented above can be exploited to design an optimization objective for deep learning, encouraging the model to distribute data densely inside its corresponding decision boundaries. To this end, in the notation of Section 2.2, let $\varphi \colon \mathcal{X} \to \mathcal{Z}$ be an encoder mapping data to a latent space $\mathcal{Z}$ equipped with a metric $d$. In practice, we usually have $\mathcal{Z} = \mathbb{R}^m$, equipped with Euclidean metric. Moreover, suppose that data is subdivided into classes, meaning that there is a partition $\mathcal{D} = \mathcal{D}_1 \sqcup \cdots \sqcup \mathcal{D}_K$, where $K$ is the number of classes and $\mathcal{D}_i$ is the data in the $i$-th class.

**Definition 2.4** ([21])**.** Given a hyperparameter $\beta \in \mathbb{R}_{>0}$, the *topological densification loss* is:

$$\mathcal{R} = \sum_{i=1}^{K} \sum_{w \in \dagger(\varphi(\mathcal{D}_i))} |w - \beta|.$$

For a Euclidean latent space $\mathcal{Z} = \mathbb{R}^m$, it can be shown that $\mathcal{R}$ is differentiable w.r.t. $\varphi(\mathcal{D}_i)$ almost everywhere, with an explicit expression for the derivatives. This allows to deploy gradient-based methods to minimize the topological densification loss over (the parameters of) $\varphi$.

We remark that the content of this section can be extended to higher-dimensional *persistent homology* [13]. The above definitions corresponds to an explicit construction of the $0$-dimensional persistent homology of the *Vietoris-Rips complex*, given that in this case all the birth times are $0$. By considering $i$-th dimensional homology, it is possible to define persistence diagrams for every $i \geq 0$ – see Appendix (Section A). The topological loss can then be extended based on features of the points in the corresponding persistence diagrams. Moreover, other filtration constructions, such as the more efficient Witness complexes [36], can replace the Vietoris Rips complex. However, following [21], we focus on the $0$-dimensional Vietoris-Rips persistent homology for simplicity, leaving further investigation of higher-dimensional extensions and other filtrations for future work.

# 3 Method

## 3.1 Robust Relative Transformation

In this section, we introduce a variation of the relative transformation that makes it invariant to the intertwiner group induced by common activation functions. This factors out non-isotropic rescalings in the relative representation, resulting in more robust model stitching. The idea behind our new relative representation boils down to introducing a Gaussian normalization with respect to a batch of data, i.e., a simple form of *batch normalization* [23] (without learnable parameters).

In the notation of Section 2.1, let $\varphi \colon \mathcal{X} \to \mathcal{Z} = \mathbb{R}^m$ be a representation and $\mathcal{A} = \{a_1, \ldots, a_k\} \subset \mathcal{Z}$ be a set of anchors. Moreover, the following definition will depend on an additional set $\mathcal{B} \subset \mathcal{Z}$, which in practice will correspond to the representation of a batch of data. For $z \in \mathcal{Z}$ we denote by $\widehat{z}^{\mathcal{B}}$ its Gaussian normalization w.r.t. $\mathcal{B}$, obtained by subtracting to $z$ the mean of $\mathcal{B}$ and by dividing each component of $z$ by the standard deviation of $\mathcal{B}$ in the corresponding direction. Technically, we assume that the standard deviations of $\mathcal{B}$ are non-vanishing, which is a generic condition.

**Definition 3.1.** The *robust relative representation* of $z \in \mathcal{Z}$ w.r.t. $\mathcal{A}$ and $\mathcal{B}$ is

$$T_{\text{rob}}^{\mathcal{B},\mathcal{A}}(z) = T_{\text{rel}}^{\widehat{\mathcal{A}}^{\mathcal{B}}}(\widehat{z}^{\mathcal{B}}).$$

Intuitively, the introduction of the normalization transforms all the components of the latent space to the same canonical scale induced by $\mathcal{B}$. Therefore, the robust version is invariant to scaled permutations and shifts, as shown by the following result.

**Proposition 3.1.** *Suppose that* sim *is the cosine similarity, and consider an $m \times m$ permutation matrix $P$, a diagonal one $D$, and a vector $h \in \mathbb{R}^m$. Denote by $\mathcal{A}', \mathcal{B}', z'$ the image of $\mathcal{A}, \mathcal{B}, z$ via $z \mapsto DPz + h$, respectively. Then:*

$$T_{\text{rob}}^{\mathcal{B},\mathcal{A}}(z) = T_{\text{rob}}^{\mathcal{B}',\mathcal{A}'}(z')$$

*Proof.* Let $\mu$ be the mean of $\mathcal{B}$ and $\Sigma$ be the diagonal matrix of its standard deviations in the corresponding components. Then, by definition, $\widehat{z}^{\mathcal{B}} = \Sigma^{-1}(z - \mu)$. The mean and standard deviations of $\mathcal{B}'$ are $DP\mu + h$ and $P\Sigma P^\top D$, respectively. Denote $\widetilde{D} = P^\top DP$. Then:

$$
\begin{aligned}
\widehat{z'}^{\mathcal{B}'} &= \widehat{DPz + h}^{\mathcal{B}'} \\
&= (P\Sigma P^\top D)^{-1}(DPz + h - (DP\mu + h)) \\
&= D^{-1}P\Sigma^{-1}P^\top DP(z - \mu) \\
&= D^{-1}P\Sigma^{-1}\widetilde{D}(z - \mu) \\
&= D^{-1}P\widetilde{D}\Sigma^{-1}(z - \mu) \qquad\qquad (\widetilde{D} \text{ and } \Sigma^{-1} \text{ are diagonal}) \\
&= P\Sigma^{-1}(z - \mu),
\end{aligned}
$$

and similarly for $\widehat{a_i'}^{\mathcal{B}'}$ for every $a_i \in \mathcal{A}$. Since the cosine similarity is invariant to linear isometries (i.e., orthogonal matrices), we obtain:

$$\text{sim}\left(\widehat{a_i'}^{\mathcal{B}'}, \widehat{z'}^{\mathcal{B}'}\right) = \text{sim}\left(\Sigma^{-1}(a_i - \mu), \Sigma^{-1}(z - \mu)\right) = \text{sim}\left(\widehat{a_i}^{\mathcal{B}}, \widehat{z}^{\mathcal{B}}\right),$$

which implies the desired claim. $\qquad\qquad\square$

As a consequence, the robust relative transformation is invariant to the intertwiner groups induced by common activation functions (e.g. GELU, ReLU, sigmoid) – see discussion after Definition 2.2. Note that this invariance property differs from the one satisfied by the original version of the relative transformation (Definition 2.1). The latter is invariant to isotropic rescalings – or, equivalently, to diagonal matrices with equal diagonal entries – and linear isometries. Therefore, our version trades off invariance to isometries other than permutations with more general non-isotropic rescalings. We claim that this tradeoff is advantageous in high dimensions. An arbitrary orthogonal matrix can be approximated by a permutation one, with the error decreasing as the dimension grows [3]. Since latent spaces of contemporary deep learning models are typically high-dimensional, the robust relative transformation approximately exhibits, to an extent, invariance to arbitrary isometries, together with the added invariance to arbitrary non-isotropic rescalings. We empirically compare the robust version with the classical one as part of our experimental investigation – see Section 4.1.

### 3.2 Topological Densification of Relative Representations

In this section, we explore various options and improvements for combining the topological densification loss (Definition 2.4) with (robust) relative representations. Since topological features of high-dimensional spaces encode semantic information [39], this additional regularization is expected to improve both the performance of the individual models, as well their zero-shot stitching capabilities, especially in low-data regimes. Moreover, to further improve the latent space similarity between models, we will apply a consistent regularization by using the same $\beta$ hyperparameter.

A fundamental choice is whether to apply the topological densification loss before or after the relative transformation. We refer to these setups as *pre-relative* and *post-relative*, respectively. Figure 3 summarizes these options, including their combination consisting of regularizing both before and after the transformation. We will evaluate and compare all of these options empirically in Section 4.

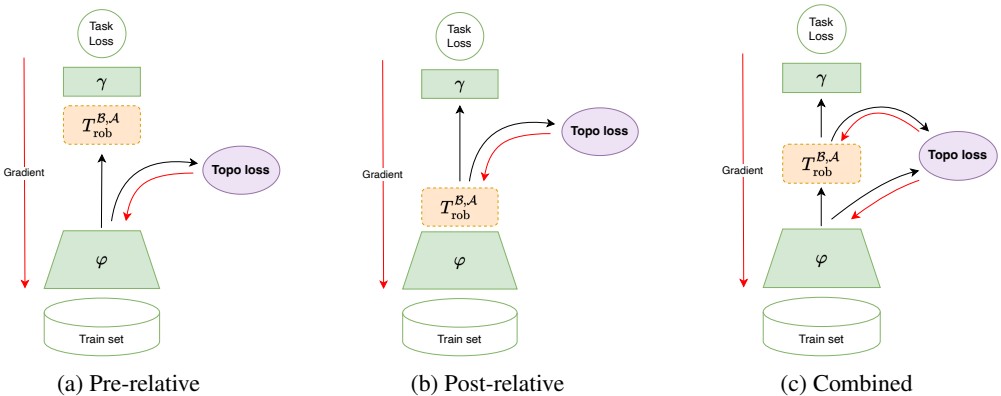

|  (a) Pre-relative  |  (b) Post-relative  |  (c) Combined  |

Figure 3: Different topological regularization setups for the relative transformation.

Additionally, we consider the problem of constructing data batches for the topological densification loss. Batching is necessary for stochastic gradient descent and, in general, for efficiency. However, topological losses are notoriously subtle to implement in a batched fashion since, intuitively, persistent homology captures global features in data spaces, which cannot be extracted from those of the batches. The original work [21] suggests constructing batches where each data class is equally represented. This is implemented by sampling (with replacement) data from the same class. However, this approach can cause conflicts with other components of the model. In particular, it is incompatible with batch normalization, which is a fundamental ingredient in our robust relative transformation. To address these challenges, we introduce a novel batch construction method. The latter consists in aggregating sub-batches sampled from class-specific datasets, with an additional sub-batch sampled from the original dataset. This results in an efficient batch construction procedure that is compatible with both the topological densification loss and batch normalization. A more detailed description is provided in the Appendix (Section B).

## 4 Experiments

In this section, we provide an empirical investigation of our proposed improvements to relative representations. To this end, we train Transformer models on a natural language task. The task is split across languages, corresponding to different domains. The cross-domain stitching procedure enables to transfer between languages, i.e., it implements a form of zero-shot translation.

**Data.** We perform a random subsampling of 1% of the Amazon Reviews dataset [24], where the task consists in predicting user ratings in a range from $1$ to $5$ stars. We consider such task in two languages: English ('en') and French ('fr'). The anchor sets $\mathcal{A}$ and $\mathcal{A}^*$ (notation from Section 2.1) consist of translated texts. Specifically, we first select a number of anchors from the English dataset equal to the dimensionality of the latent space (768 dimensions), and then translate them to other languages using Google Translate. A Python code implementing our models and experiments is available at a public anonymized repository: `https://tinyurl.com/TopoRelTrans`.

**Models and Training.** We use pre-trained RoBERTa models [30] as the base for our representation maps $\varphi$, and a single layer on top of these representations for the head $\gamma$. The models are fine-tuned end-to-end on the task described above via two approaches: the 'Absolute' case, where no relative transformation is applied during training, and the 'Relative' case, where the relative transformation is employed, as described at the end of Section 2.1. We train the models via the AdamW optimizer [31] with a layer-wise learning rate decay on the parameters of $\varphi$. The initial learning rate is $3.5 \cdot 10^{-5}$ and decay rate of $0.65$. The learning rate of $\gamma$ is $2 \cdot 10^{-4}$. Training is performed over 40 epochs, with batch size of 16 and gradient accumulation every 6 steps. Additionally, we use a linear cyclic scheduler for the weights of the topological densification loss, which is a common strategy for optimizing combined objectives [16]. For the relative case, we update the 768 anchor embeddings every 500 optimization steps. This reduces the computational cost, and has minor effects on training stability due to the learning rate schedule for $\varphi$.

**Evaluation Metrics.** The performance of the models is evaluated using three metrics: the Mean Absolute Error (MAE) between user ratings seen as integers, the $F_1$ classification score, and the classification accuracy (Acc). All the scores are multiplied by 100 for better readability. As depicted in Figure 1, to test the stitching performance, we match the representation network's language with the test dataset's language, while the classification head is drawn from a network trained in the other language.

Our experimental setup is similar to the original work [35]. However, in order to test the models in a more challenging fine-tuning scenario, we train our models on fewer data (only 1% of the Amazon Reviews dataset). Moreover, we deploy a simpler linear head $\gamma$, instead of a deeper network. This naturally results in worse reported performance as compared to the original work.

## 4.1 Comparison of Relative Transformations

In our first experiment, we compare the robust relative transformation (Section 3.1) with the original one, showcasing improved performance on the considered cross-domain stitching scenario. To this end, we experiment with both the original relative transformation ('Relative Vanilla') and our robust version ('Relative Robust'), together with the baseline ('Absolute') where no transformation is deployed.

Table 1: Performance comparison on zero-shot model stitching.

| $\gamma$ | $\varphi$ | Absolute | | | Relative Vanilla | | | Relative Robust | | |
|---|---|---|---|---|---|---|---|---|---|---|
| | | Acc ($\uparrow$) | $F_1$ ($\uparrow$) | MAE ($\downarrow$) | Acc ($\uparrow$) | $F_1$ ($\uparrow$) | MAE ($\downarrow$) | Acc ($\uparrow$) | $F_1$ ($\uparrow$) | MAE ($\downarrow$) |
| en | en | $59.26_{\pm0.66}$ | $58.27_{\pm0.83}$ | $49.52_{\pm0.89}$ | $38.84_{\pm1.23}$ | $23.50_{\pm2.77}$ | $84.95_{\pm9.48}$ | $60.84_{\pm0.64}$ | $60.30_{\pm0.72}$ | $45.35_{\pm0.74}$ |
| | fr | $24.28_{\pm10.11}$ | $22.27_{\pm8.86}$ | $139.27_{\pm35.32}$ | $40.96_{\pm2.40}$ | $31.15_{\pm3.29}$ | $73.09_{\pm5.18}$ | $49.92_{\pm1.51}$ | $50.13_{\pm1.60}$ | $57.56_{\pm1.60}$ |
| fr | en | $24.96_{\pm9.27}$ | $23.19_{\pm8.12}$ | $132.35_{\pm24.01}$ | $35.42_{\pm1.16}$ | $20.86_{\pm1.09}$ | $79.68_{\pm11.68}$ | $60.74_{\pm0.88}$ | $60.18_{\pm1.14}$ | $45.19_{\pm1.16}$ |
| | fr | $49.26_{\pm1.04}$ | $48.74_{\pm0.73}$ | $63.89_{\pm1.50}$ | $41.99_{\pm3.18}$ | $35.33_{\pm4.55}$ | $67.77_{\pm2.24}$ | $50.31_{\pm0.88}$ | $50.95_{\pm0.82}$ | $57.08_{\pm1.22}$ |

Table 1 reports the results in terms of mean and standard deviation for 5 experimental runs. On the cross-domain setup – i.e., $\gamma =$ en and $\varphi =$ fr, and vice versa – our robust version significantly outperforms the original one on all the metrics considered. Specifically, when stitching from English to French, accuracy and $F_1$ increase by around 9 and 19 respectively, while MAE decreases by 15. The performance gain is even more drastic when stitching from French to English, with an increase of 25 and 40 in $F_1$ and accuracy, and a decrease of 34 in MAE. The larger improvements in the latter setup can be explained by the fact that the RoBERTa models are pre-trained better in the English language. This results in fine-tuned representations $\varphi$ whose extracted features are, generally speaking, more transferable, mitigating the effect of the stitching technique deployed. The explanation is confirmed by the fact that the cross-domain performance without relative representations (Absolute column in the table) is better in the English-to-French setup than vice versa. Lastly, we remark that, surprisingly, the robust relative transformation exhibits slightly improved performance with respect to the Absolute version when the domain is not changed, i.e. when both $\gamma$ and $\varphi$ are set to either fr or en. This does not happen for the non-robust transformation. We conclude that not only our proposed transformation is advantageous to a large extent for transferring across domains, but the model can benefit from it even for the original task on low data regimes.

## 4.2 Analysis of Topological Densification

In this section, we assess the benefits of the topological densification loss in the context of relative representations, as described in Section 3.2. In all the following experiments, we deploy our robust version of the relative transformation, which has been assessed in the previous section.

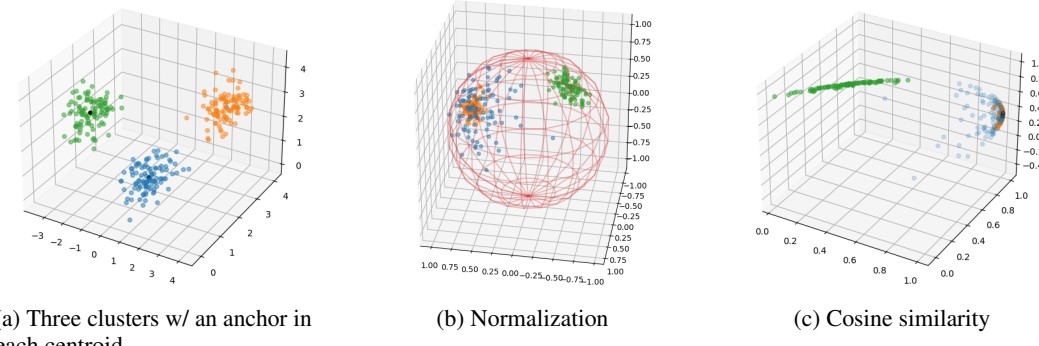

(a) Three clusters w/ an anchor in each centroid

(b) Normalization

(c) Cosine similarity

Figure 4: Non-cluster-preserving relative transformation example

We start by comparing the three possible approaches for implementing the topological densification – see Figure 3. Initially, we have investigated the pre-relative and post-relative options which, however, had negative impact on the performance. For the pre-relative case, a possible explanation is that the relative transformation can fail to preserve the topology of latent data. An example of this is presented in Figure 4, where the representation consists of three clusters, two of which possess (approximately) collinear centroids. These two clusters are merged by the relative transformation, becoming indistinguishable. For the post-relative case, we hypothesize that the model discovers an anchor configuration that yields tighter clusters in the post-relative space, while preserving their spread in the pre-relative space. This

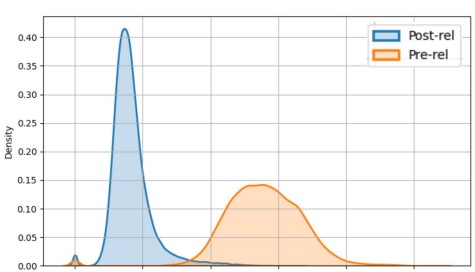

Figure 5: Distribution of death times when training with post-relative topological densification on the English dataset.

is confirmed by examining the distribution of death times (see Figure 5), which exhibit a significantly smaller mean in the post-relative space than in the pre-relative one. We believe this configuration can lead to information bottlenecks, because compressing the clusters with this nonlinear transformation might cause a loss of expressiveness in the latent space.

We conclude that the best option is deploying the two topological regularization losses simultaneously, which we implement via a linear combination of them with two weights $\lambda_1$ (pre-relative) and $\lambda_2$ (post-relative). Through an extensive hyperparameter search, we discovered the optimal values of $\lambda_1 = 2 \times 10^{-3}$ and $\lambda_2 = 1.8 \times 10^{-2}$, along with a topological densification parameter $\beta = 3$ for both the French and English datasets. With this setup, the pre-relative and post-relative distributions of death times overlap – see Figure 6. This addresses the above-mentioned challenge, and indicates that, with this setup, the relative transformation preserves the topological information.

Table 2 reports the performance scores when fine-tuning with topological regularization for 5 experimental runs. As compared to Table 1, a slight improvement is evident in the cross-domain context. Specifically, for the English-to-French setup, all the scores improve by around $0.5$, while they improve by around $1$ in the French-to-English setup. A similar increase can be observed when the domain is not changed. This shows that topological densification is beneficial for transferring between domains. However, it comes with a higher computational cost, as discussed in Section 2.3. This additional cost concerns only the training phase, while the zero-shot transfer procedure is unaffected.

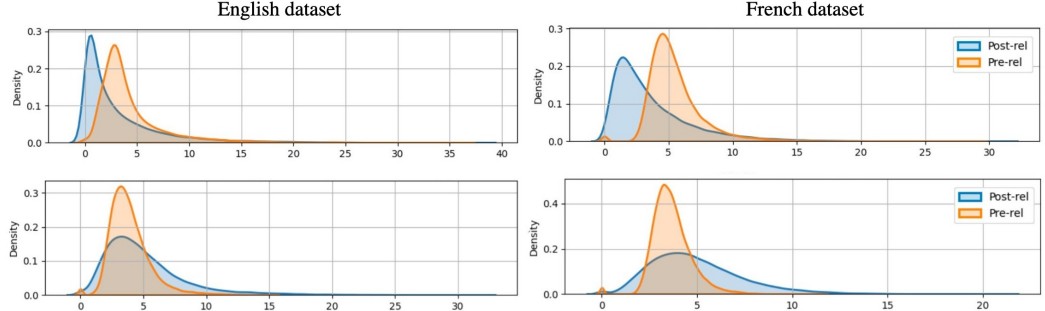

Figure 6: Distribution of death times on the English (left) and French (right) datasets. **Top**: without topological densification. **Bottom**: with a combination of pre-relative and post-relative topological densification.

Table 2: Performance with topological densification.

| $\gamma$ | $\varphi$ | Relative Robust | | |
| | | Acc ($\uparrow$) | $F_1$ ($\uparrow$) | MAE ($\downarrow$) |
| --- | --- | --- | --- | --- |
| en | en | $61.16_{\pm 0.42}$ | $61.26_{\pm 0.18}$ | $44.63_{\pm 0.26}$ |
| | fr | $50.48_{\pm 1.04}$ | $50.85_{\pm 1.25}$ | $57.70_{\pm 0.73}$ |
| fr | en | $60.93_{\pm 0.56}$ | $61.23_{\pm 0.46}$ | $44.54_{\pm 0.51}$ |
| | fr | $50.63_{\pm 0.79}$ | $50.97_{\pm 0.85}$ | $57.76_{\pm 0.71}$ |

## 5 Conclusions, Limitations, and Future Work

In this work, we have introduced two improvements to relative representations. These improvements consist in a novel normalized version of the relative transformation, and in the deployment of a topological regularization loss in the fine-tuning procedure. We have investigated empirically our proposals on a natural language task, showcasing improved performance as compared to the original relative representations.

Our robust relative transformation is invariant to non-isotropic rescalings and permutations, which are the only symmetries of common activation functions. However, the original relative transformation is invariant to all the isometries (and isotropic rescalings) of the latent space. Even though trading off non-permutational isometries with non-isotropic rescalings is advantageous in high dimensions (see discussion after Definition 3.1), the challenge of designing a robust relative transformation that is invariant to all isometries and all rescalings remains open. This would lead to an even more robust zero-shot model stitching procedure, and therefore represents a fundamental challenge for future research.

Another future direction from the topological perspective is exploring topological regularization procedures beyond densification. This includes extensions of the latter to higher-dimensional persistent homology, as discussed at the end of Section 2.3 and in Section A, or alternative regularization losses – see [19] for an overview. The core challenges behind scaling up such regularizers are their computational complexity and the difficulty of batching. Yet, they might result in advantages for either general or specific domains when applying model stitching techniques.

## Acknowledgements

This work was partially supported by the Wallenberg AI, Autonomous Systems and Software Program (WASP) funded by the Knut and Alice Wallenberg Foundation. Alejandro García Castellanos

is funded by the Hybrid Intelligence Center, a 10-year programme funded through the research programme Gravitation which is (partly) financed by the Dutch Research Council (NWO).

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

# A   Extended Topological Densification

In Section 2.3 we describe 0-th persistent homology in relation to topological densification [21]. We now present a concise definition of $n$-th persistent homology for $n \geq 0$, and elaborate on possible generalizations of the topological densification loss (Definition 2.4). For an introduction to persistent homology and topological data analysis we refer the reader to [8].

The persistent homology pipeline can be subdivided in steps:

**From Data to Geometry.** The first step consists in associating a nested sequence (i.e., a filtration) of simplicial complexes to the data. Simplicial complexes are simple geometric objects that generalize the notion of a graph and can be described combinatorially. If data is in the form of a finite metric space $\mathcal{D}$ the filtration of simplicial complexes can be for example constructed by computing the Vietoris-Rips complex, the Cech complex or the Witness complex at scale $\varepsilon$, for increasing values of the parameter $\varepsilon \in \mathbb{R}$.

**From Geometry to Algebra.** The topology of a simplicial complex and in particular its homology can be computed in an algorithmic way through simplicial homology. Fixed a natural number $n$ and a coefficient field, the $n$-th homology of a simplicial complex is a vector space encoding connectivity of the simplicial complex. The 0-th homology describes connected components, 1-st homology describes cycles that are not bounded by faces, 2-nd homology describes voids and for $n \geq 2$ one can think of higher dimensional analogues. In the cases of a sequence of simplicial complexes, $n$-th homology can be applied to the whole sequence to obtain a sequence of vector spaces and linear maps. This is referred to $n$-th persistent homology module.

**Representing Persistent Homology.** Persistent homology modules are well known and simple algebraic objects, due for example to the fact that the sequence of vector spaces constituting a persistence module is indexed by $\mathbb{R}$ and that it is essentially discrete. Due to Gabriel's theorem, or equivalently the decomposition theorem of modules over a Principle Ideal Domain, persistent homology modules admit a barcode decomposition, they can be decomposed as sum of addends called bars. A bar is completely described by a pair of real numbers: the birth and the death of the bar. The collection of all birth and death pairs of the bars in a barcode decomposition of a persistence module is the so called persistence diagram. Persistence diagrams and their features are used both as data descriptors and for improving data representations in deep learning, for example through regularization.

As in Definition 2.4 we denote by $\mathcal{D}$ the dataset, by $\varphi \colon \mathcal{X} \to \mathcal{Z}$ an encoder mapping data to a latent space, and assume the data is partitioned into classes $\mathcal{D} = \mathcal{D}_1 \sqcup \cdots \sqcup \mathcal{D}_K$. The topological densification loss is defined as

$$\mathcal{R} = \sum_{i=1}^{K} \sum_{w \in \dagger(\varphi(\mathcal{D}_i))} |w - \beta|.$$

where $\dagger(\varphi(\mathcal{D}_i))$ is the set of death times in the persistence diagram of $\varphi(D_i)$. For 0-th persistence, death times correspond to lengths of bars in a barcode, since all bars start at parameter value 0. Consider now the case of $n$-th persistent homology of the $\varphi(\mathcal{D}_i)$ and the corresponding persistence diagrams $\ddagger(\varphi(\mathcal{D}_i))$. The formula for of the topological densification loss can be generalized as:

$$\mathcal{R} = \sum_{i=1}^{K} \sum_{(a,b) \in \ddagger(\varphi(\mathcal{D}_i))} |\ell(a, b) - \beta|.$$

where $l$ is a real valued function assigning a weight to each point in the persistence diagram. Examples of such functions are the length of a bar $\ell(a, b) = b - a$ or the lifespan of a bar [1] – a generalization of length defined as $\ell_F(a, b) = F(b) - F(a)$, where $F \colon \mathbb{R} \to \mathbb{R}$ is an increasing bijection.

# B   Batch Construction

As mentioned in Section 3.2, the original work on topological densification [21] proposes the following batch construction. A batch consists of $b$ sub-batches, where each sub-batch contains $n$ samples from the same class.

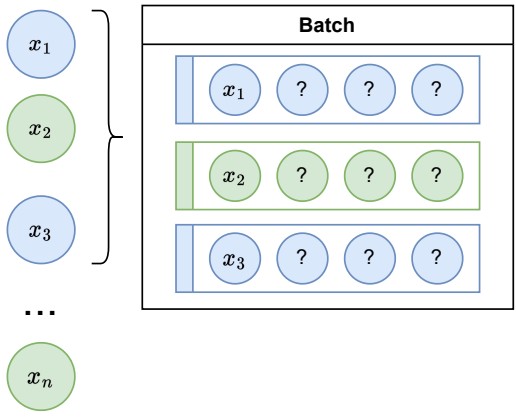

Figure 7: The original dataloader for topological densification.

More specifically, the dataloader in the original work is implemented as follows (see Figure 7 for an illustration):

1. A batch of size $b$ is sampled from the original dataset.
2. For each datapoint $x$ in the batch, a sub-batch is constructed by sampling with replacement $n - 1$ more datapoints with the same class as $x$.

This approach comes with two major issues. Firstly, in the presence of significant class imbalance, classes are likely to be under-represented within batches. This can potentially lead to catastrophic forgetting and unstable training. Secondly, this method is computationally intensive, since training requires processing the original dataset $n$ times in a single epoch.

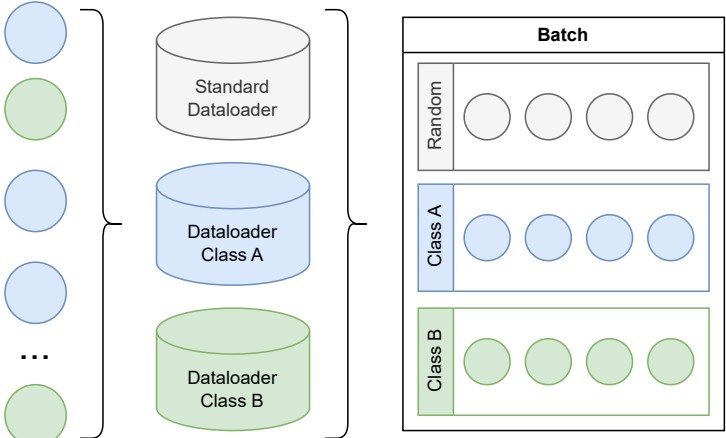

Figure 8: Our dataloader for topological densification.

We propose a new dataloader that follows the same high-level principle, but addresses the above concerns. Our dataloader is implemented as follows (see Figure 8 for an illustration):

1. For each class, we create a separate dataloader containing all the data from that class. Additionally, we create a standard dataloader containing all the data.
2. We aggregate $n$ samples from each dataloader, resulting in a batch consisting of $K + 1$ sub-batches, where $K$ is the number of classes.

This new dataloader preserves the class structure necessary for applying the topological regularization loss. Its computational overhead is comparable to training directly with the standard dataloader. Moreover, it prevents class imbalance, since each class is guaranteed to be represented at least $n$ times within a batch.

The introduction of the sub-batch from the standard dataloader is relevant for the normalization procedure involved in our robust relative transformation. Namely, we use the (encoding via $\varphi$ of the) sub-batch from the standard dataloader as $\mathcal{B}$ (notation from Section 3.1) when fine-tuning robust relative representations. This is crucial, for example, when the original dataset exhibits class imbalance. In that case, the normalization should be performed with a set $\mathcal{B}$ that is representative of the original dataset – which is addressed by the sub-batch from the latter – while the topological loss still benefits from class-balanced data – which is addresses by the other class-specific sub-batches.

