# OpenReview forum: "Relative Representations: Topological and Geometric Perspectives"
_NeurIPS.cc/2024/Workshop/UniReps — UniReps_

### Official Review · Reviewer_K13q · 2024-10-05
**Two modifications to the relative representations method**

**Rating:** 6
**Confidence:** 3

**Review:**

The paper introduces two modifications to the relative representation approach.

The first modification introduces a transformation from absolute to relative representations that is invariant to combinations of permutations, element-wise rescalings, and shifts (eq. at line 190). In contrast, the transformation proposed by Moschella et al. (2023) is invariant to linear isometries and global rescalings (eq. at line 77). The paper’s readability could be improved by comparing these equations side by side or numbering them, which would make it easier for readers to compare the two approaches.

The original and modified methods are invariant to different, though overlapping, classes of transformations, making it an empirical question which will perform better. The authors argue that their proposed transformation is nearly as expressive as the original in high dimensions. Experiments suggest that the new transformation performs better, at least for the task considered.

I found the explanation in lines 199-208 helpful. In my view, the significance of the proposed approach would have been clearer if this motivation had been introduced earlier in the paper. Without it, the reasoning behind this modification of the relative representations approach is unclear, especially since the paper begins with statements such as:
> For example, there is extensive empirical evidence that the representations extracted by neural networks are isometric

which seems to vindicate the original approach. Although the authors provide some motivation for their modification in lines 35-43, it's unclear how these statements align with one another.

The motivation for the second modification, fine-tuning based on topological densification, was not entirely clear to me. Unlike the first modification, it's unclear what class of transformations this fine-tuning is invariant to. In some passages, the authors refer to "decision boundaries", even though their method is meant to work beyond supervised settings. However, the empirical investigation of their hypothesis could still be of interest.

Overall, the paper introduces valuable additions to the relative representations method, which are relevant to the workshop's theme. However, the two contributions feel somewhat disconnected, and the readability of some sections could be improved.

---

### Official Review · Reviewer_Dv2o · 2024-10-06
**Very well written paper. All sections are clear and concise. The topics fits very well into the scope of this workshop. The paper fails to describe the vanilla relative representation, which has implications on the validity of the results.**

**Rating:** 6
**Confidence:** 3

**Review:**

The paper proposed two mechanisms to improve Relative Representations for model stitching: Normalization and Densification. The paper has a nice and clear theoretical motivation behind the proposed mechanisms.

1) Normalization suggests a batch normalization layer to become invariant to non-isometric scaling. In 4.1, the authors report the results, comparing the Robust Rel. Repr. to a Vanilla framework, using cosine distance as a similarity measure. It is unclear whether the Vanilla framework includes centering the latent space. If e.g. the latent space is only in the positive domain and far away from 0, all angles will be small. Thus, it is not clear whether the reported improvement comes from the centering or the scaling part of the batch norm.

2) Densification: this improvement is rather small compared to reported standard deviations. I wonder whether other similarity measures e.g. euclidean distance, will benefit more from densification, as projections onto the unit sphere in the cosine similarity will lead to a densification of the latent space.

Even though the baseline method is not entirely clear, the results and theoretical motivation of the proposed methods provide us with a better understanding of the Relative Representation framework. I therefore rate the paper as above acceptance.

---

### Official Review · Reviewer_6dy5 · 2024-10-06
**Relative Robust Representation with Batch Normalization and Topological Loss**

**Rating:** 4
**Confidence:** 4

**Review:**

The paper introduces a new variant of relative representation, termed “Relative Robust,” by incorporating batch normalization before the relative representation computation and adding a topological loss both before and after the representation step. The proposed method is tested on the task of zero-shot stitching, showing improved performance over standard relative representations.

The theoretical framework is sound, and the explanation of the method is clear and solid. However, there are notable concerns in the experimental section:

	1.	Inconsistent Baseline Performance: The reported performance for vanilla relative representation (particularly in en-en and fr-fr tasks) shows a drastic drop, which is inconsistent with the results from the original paper. This raises concerns about whether the method was correctly implemented. Stronger justification for these results is necessary, as they currently undermine the credibility of the proposed method’s improvement.
	2.	Limited Dataset Reproduction: The experiments seem narrow in scope, especially when compared to the original dataset used in prior work. To reinforce the validity of the results, it would be beneficial to reproduce more experiments from the original dataset, including additional datasets and languages as tested in the original study.
	3.	Topological Loss Efficacy: While the introduction of topological loss is interesting, its contribution to performance improvement is marginal when compared to batch normalization alone. This raises the question of whether the additional computational complexity of topological loss is justified by the limited performance gains. Further analysis of the added computation time and whether it meaningfully enhances the results would be valuable.
	4.	Presentation of Results:
	•	Figures 5 and 6 would benefit from clearer labeling, especially the x-axis.
	•	Tables 1 and 2 could be combined for better readability and ease of comparison.

Overall, while the core idea of improving relative representation through batch normalization and topological loss is intriguing, the current experimental results are not fully convincing. Strengthening the experimental section, particularly with respect to reproducibility and addressing the suspicious baseline results, will be critical to enforce the paper’s claims.

---

### Official Review · Reviewer_zsHe · 2024-10-06
**Novel application of geometry and topology to improving relative representations**

**Rating:** 6
**Confidence:** 3

**Review:**

The paper presents two improvements to relative representations, a method used in zero-shot model stitching: a normalization procedure that enhances invariance to non-isotropic rescalings and permutations, and the introduction of topological densification as a regularization technique to improve the robustness of model representations. The work aims to enhance model performance in cross-domain tasks, demonstrated through natural language experiments where models are fine-tuned for zero-shot translation between English and French. Both improvements are evaluated empirically and shown to improve stitching performance in low-data regimes.

### Pros
- The application of topological data analysis to relative representations is innovative and pushes the boundary of model stitching and zero-shot learning. Additionally, the idea of creating representations invariant to the intertwined groups aligns well with recent trends in invariant representation learning. These contributions are novel within the scope of zero-shot learning.
- The empirical results clearly demonstrate performance improvements over the baseline. The improvements in performance suggest that the techniques could be broadly applicable to many tasks requiring cross-model or cross-domain representation transfer.

### Cons
- The clarity of the paper is mixed. The introduction and motivations are well articulated, but the technical depth and mathematical rigor may overwhelm readers without a background in topology or geometric deep learning. The figures are helpful though.
- The practical significance is limited by the computational cost of topological regularization, which may hinder adoption in large-scale models or real-time applications.

---

### Author Response · Authors · 2024-10-17

We are grateful to all the reviewers for their comments and feedback, which we have incorporated into our paper.

A major concern was raised regarding our reported results for the baseline, which are significantly lower than the ones in the original work [1]. We wish to clarify that this is due to the fact that we drastically subsample the data from Amazon Reviews – a large-scale natural language dataset. We only use 1% of the training data for fine-tuning, as mentioned in Section 4 (paragraph ‘Data’). Moreover, we use a simple linear layer as a classification head $\gamma$, as mentioned in Section 4 (paragraph ‘Models and Training’), instead of a deep network as in the original work. This was done not only for compute reasons, but to also test the models on a more challenging fine-tuning scenario, leading to lower performance scores. We acknowledge that these differences are important to clarify explicitly –  we have included an additional paragraph in Section 4 to this end.


[1] Moschella et al., Relative Representations Enable Zero-Shot Latent Space Communication, ICLR 2023.

---

### Decision · Program_Chairs · 2024-10-10

**Decision:**

Accept

**Comment:**

In light of the reviewers' feedback and relevancy of the submission, we are pleased to accept this paper for presentation at UniReps 2024. We kindly ask the authors to incorporate the reviewers' suggestions and feedback in the final camera-ready version of the manuscript, especially regarding the baseline consistency.